Mitogenomic phylogeny of Tetrigoidea (Insecta, Orthoptera), with a focus on the genus Zhengitettix

Li Xuejuan
Dou Wenli
Lin Liliang ll_lin@163.com
Shaanxi Normal University , Xi’an , China
Sunny Armando
Electronic publication date: 2025 Jun 9
Publication date: 2025
Volume: 13
Electronic Location ID: e19521
Received 2025 Jan 27; Accepted 2025 May 5
Copyright: ©2025 Li et al.
Copyright year: 2025
Copyright holder: Li et al.
License: This is an open access article distributed under the terms of the Creative Commons Attribution License, which permits unrestricted use, distribution, reproduction and adaptation in any medium and for any purpose provided that it is properly attributed. For attribution, the original author(s), title, publication source (PeerJ) and either DOI or URL of the article must be cited.
License URL: https://creativecommons.org/licenses/by/4.0/

Keywords: Tetrigoidea, Mitogenome, Genome feature, Phylogenetic analysis, Divergence time

Funding: National Natural Science Foundation of China 31801993 Fundamental Research Funds for the Central Universities, China GK202304021 Postdoctoral Science Foundation of Shaanxi Province, China 2017BSHEDZZ99 This work was supported by the National Natural Science Foundation of China (Grant No. 31801993), Fundamental Research Funds for the Central Universities, China (Grant No. GK202304021), Postdoctoral Science Foundation of Shaanxi Province, China (Grant No. 2017BSHEDZZ99). The funders had no role in study design, data collection and analysis, decision to publish, or preparation of the manuscript.

==============================
The mitochondrial genome (mitogenome) has been widely used to infer the phylogeny, origin and evolution of Orthopteran insects. Although several mitogenomic data have been used to study the phylogenetic relationships of Tetrigoidea (Orthoptera), the phylogenetic status of several subfamilies and tribes was still unclear due to the limited sampling of taxon. To further analyze the mitogenomic features and phylogeny of Tetrigoidea, five mitogenomes (Zhengitettix curvispinus, Z. hainanensis, Scelimena melli, Eucriotettix oculatus and Thoradonta yunnana) were sequenced and analyzed in this study, with Z. hainanensis being the newly published mitogenome and Z. curvispinus and S. melli being the complete mitogenomes. Nucleotide composition showed that more A and T bases than C and G bases were found in the sampled mitogenomes, with A- and C-skew. A large intergenic region containing tandem repeats was identified between trnS(ucn) and nad1 in the Z. curvispinus mitogenome. The protein-coding genes (PCGs) used ATG and TAA as the most common initiation and termination codons, respectively. The tRNAs showed a typical clover secondary structure in the Z. curvispinus. The A+T-rich region contained tandem repeats in Z. curvispinus. Phylogenetic analyses of Tetrigoidea based on the maximum likelihood (ML) and Bayesian inference (BI) method supported several non-monophyly subfamilies and tribes, such as Scelimeninae and Thoradontini. Divergence time results showed that Tetrigoidea is one of the basal branch of Orthoptera, with Batrachideinae splitting first, followed by Tripetalocerinae. The genera diverged over a relatively long period, expanding from the Jurassic to the Neogene. These results provide useful data for the study of the mitogenome characteristic of the Tetrigoidea and even the whole Orthoptera, and were basic resources for their phylogeny and evolution study.

Introduction

Tetrigoidea insects, the pygmy grasshoppers, that are considered to be agricultural pests, are widespread throughout the world. Species of Tetrigoidea form a unique family (Tetrigidae), belonging to Caelifera, Orthoptera. Tetrigidae contains seven subfamilies (Batrachideinae, Cladonotinae, Lophotettiginae, Metrodorinae, Scelimeninae, Tetriginae, Tripetalocerinae) and two tribes (Criotettigini, Thoradontini) (Orthoptera Species File v.5.0, Cigliano et al., 2021).

Animal mitochondrial genomes (mitogenomes), typically ∼16 kb in size, contain 37 genes, including 13 protein-coding genes (PCGs), two ribosomal RNA genes (rRNAs) and 22 transfer RNA genes (tRNAs) (Boore, 1999), and are considered ideal molecular makers for phylogenetic studies. Mitogenomes have been widely used to study mitogenome sequence feature, character evolution, phylogenetic relationship, and divergence time of Orthopteran insects (Song et al., 2015; Chang et al., 2020). For Caelifera, the mitogenome composition and gene order were relatively stable (Zhang et al., 2013; Chang et al., 2020). Tridactyloidea, located at the basal position of Caelifera, and the gene order of cox2-trnK-trnD-atp8 were found in all three families of Tridactyloidea mitogenomes, which order retained the ancestral state, while the gene arrangement with cox2-trnD-trnK-atp8 order was identified in the remaining Caeliferan superfamily mitogenomes (Song et al., 2015). This rearrangement evolved in the common ancestor of Acrididea (Acridoidea, Caelifera) and was estimated to have occurred in the Late Permian or Early Triassic (Song et al., 2015).

Most previous studies related to Tetrigoidea species focused on their morphological research (Deng, 2016), and corresponding molecular data were limited. Studies of Tetrigoidea species based on molecular data mainly focused on mitochondrial genes and whole mitogenomes (Lin et al., 2015; Lin et al., 2017; Li et al., 2021; Li, Liu & Lin, 2021; Luo, Zhang & Deng, 2024) and transcriptome (Qiu et al., 2017; Liu, Li & Lin, 2023). In recent years, high-throughput sequencing technology has been widely used in to obtain mitogenomes of Tetrigoidea species, the data of which have been used to study their phylogeny and evolution (Li, Liu & Lin, 2021; Guan, Huang & Deng, 2024; Luo, Zhang & Deng, 2024). In addition, two Tetrigoidea genomes have so far been assembled at the chromosome-level, including Zhengitettix transpicula (Guan et al., 2024) and Eucriotettix oculatus (Li et al., 2024), also benefiting from advances in high-throughput sequencing technology, and providing valuable genome references for further studies of their evolution. In addition, based on the mitogenome analysis results, the taxonomic status of some Tetrigoidea species was confirmed. For example, based on mitogenome sequences, Li, Liu & Lin (2021) found that Systolederus (Metrodorinae) and Teredorus (Tetriginae) clustered together and had a close phylogenetic relationship, suggesting that these two genera could probably be considered as one genus of Teredorus.

Although mitogenomes of Tetrigoidea species have been widely used to explore the phylogeny of these taxa, there were still some issues to be resolved, including the monophyly of some subfamilies and genera, and the phylogenetic relationship among different subfamilies and species. For example, the monophyly of Scelimeninae was not recovered using mitogenomic PCGs (Li et al., 2021), in which Z. curvispinus did not cluster with the other three members of Scelimeninae. Also based on the mitogenomic dataset, Systolederus spicupennis had a closer relationship with Teredorus hainanensis, breaking the monophyly of Teredorus (Li, Liu & Lin, 2021). In addition, several mitogenomic studies of Tetrigoidea species focused only on their phylogeny and less on origin and divergence, which limited the systematic evolution studies of these taxon. Therefore, the mitogenomic data, especially with the sequencing of larger samples, will be beneficial to further explore the genome characteristics, phylogenetic relationships and evolution of Tetrigoidea species.

In this study, the mitogenomes of five Tetrigoidea species, including Z. curvispinus, Z. hainanensis, Scelimena melli, E. oculatus and Thoradonta yunnana, were sequenced, assembled and annotated. All these five species belonged to Scelimenidae (Zheng, 2005), while three species (Z. curvispinus, Z. hainanensis, Scelimena melli) were classified in Scelimeninae and two species (E. oculatus and T. yunnana) were classified in Thoradontini based on Orthoptera Species File (v.5.0). Molecular genetic analysis was therefore used to verify their classification and phylogenetic relationship. Other Tetrigoidea mitogenome sequences from the GenBank database were combined, the genome features were analyzed, and phylogenetic relationships and divergence times were determined in our study. The results provide available Tetrigoidea mitogenomic resources, a new view on the phylogenetic implications of Tetrigoidea, and references for the traditional classification of this taxon.

Materials and Methods

Sampling and sequencing

Sampling information for this study is shown, with Z. curvispinus, S. melli and E. oculatus collected in Shiwandashan, Guangxi, China in 2012, Z. hainanensis collected in Qiongzhong, Hainan, China in 2017 and T. yunnana collected in Pu’er, Yunnan, China in 2007. The specimens were preserved in alcohol, stored at −20 °C and deposited in the Zoological and Botanical Museum, Shaanxi Normal University, China. Tissues without intestines from the whole specimens were used for DNA extraction and sequencing. Genomic DNA was extracted using a Tiangen DP802 kit (Tiangen, Beijing, China) with a total of 40–45 ul for each DNA extraction sample, and the quality was checked using the Qubit Fluorometer (Qubit, Paris, France). Total DNA was fragmented using an ultrasonic-mechanical method with an ultrasonic instrument (Bioruptor UCD-600 (NGS)), the fragmented DNA was used to prepare a small inserted fragment library using the VAHTSTM Fg DNA Library Prep Kit for Illumina (Illumina, San Diego, CA, USA) and the data were sequenced using the Illumina X-plus platform with a 150 bp paired-end strategy.

Assembly, annotation and feature analysis

Raw data were checked and filtered using fastp (Chen et al., 2018a), low-quality reads were filtered using fastq-filter, and assembled using the NOVOPlasty v4.3.1 program (Dierckxsens, Mardulyn & Smits, 2017) with mitogenomic sequences from Tetrix japonica (NC_018543) and closely related species [including Z. curvispinus (MT162544) and T. yunnana (NC_071832)] as seed and reference sequences. For Z. curvispinus, the raw data after trimming were used directly for mitogenome assembly, while the assembly data of the other four Tetrigoidea were a subset of the genome survey sequencing data. The assembled mitogenomes were first annotated in the MITOS WebServer (Bernt et al., 2013) and then compared with other available Tetrigoidea mitogenomic annotation information for verification. Among them, the position and secondary structure of tRNAs were referenced in comparison with other Tetrigoidea mitogenomes (Lin et al., 2017). Nucleotide composition, relative synonymous codon usage (RSCU) and genetic distance were calculated in MEGA v11 (Tamura, Stecher & Kumar, 2021). Base skew was calculated using the formula AT-skew = [(A−T)/(A+T)] and GC-skew [(G−C)/(G+C)] (Perna & Kocher, 1995). Tandem repeats were predicted using Tandem Repeat Finder with default settings (Benson, 1999).

Phylogenetic inference and divergence time

A total of 52 mitogenomic sequences were used for phylogenetic analyses, including two Tridactyloidea outgroups (Ellipes minuta and Mirhipipteryx andensis) (Table S1). Three mitogenomic datasets were used for phylogenetic analyses, including 13 PCGs, 13 PCGs plus two rRNAs (PCG_rRNA), 13 PCGs, two rRNAs plus 22 tRNAs (PCG_rRNA_tRNA). The mitogenomic sequences for each gene were individually aligned using the Muscle program in MEGA v.11 (Tamura, Stecher & Kumar, 2021). Firstly, stop codon of each PCG was removed, including incomplete the stop codon. Then, the nucleotide sequences were translated into amino acids and aligned. At last, the aligned amino acid sequences were translated back to the corresponding nucleotide sequences. SequenceMatrix v1.7.8 (Vaidya, Lohman & Meier, 2011) was used to concatenate the mitogenomic datasets. For some partial mitogenome sequences, the lacked gene sequences were defined as missing data in the combined datasets. The saturation test of the three datasets was analyzed using DAMBE v7.2.152 (Xia, 2018), with the F84 distance.

Phylogenetic analyses were performed using maximum likelihood (ML) and Bayesian inference (BI) methods. The best-fitting model was estimated using the ModelFinder program in PhyloSuite v1.2.3 (Zhang et al., 2020) under the Bayesian information criterion (BIC). The GTR+F+I+I+R4 model was used to reconstruct the ML tree for each dataset, while the GTR+F+I+G4 model was used for the BI tree. ML analyses were conducted in IQ-TREE v1.6.12 (Nguyen et al., 2015) with 1,000 bootstrap replicates. BI analyses were performed in MrBayes v3.2.7 (Ronquist et al., 2012), with two independent runs of four simultaneous Markov chains. The 10 million generations were run, sampling every 100 trees. The first 25% of trees were discarded as burn-in, and the remaining trees were used to obtain the consensus tree. Effective sample size (ESS) values were estimated in Tracer v1.5 (Rambaut, Suchard & Drummond, 2004) to ensure that the ESS value was greater than 200. Phylogenetic trees were modified using the iTOL website (https://itol.embl.de/).

Divergence times of Tetrigoidea species were estimated using the MCMCTREE program in PAML v4.9 (Yang, 1997). The PCG dataset and corresponding ML tree were used, and four records from the TimeTree website (Kumar et al., 2017) were used as calibration points, including 4.4−8.0 Mya between T. japonica and Alulatettix yunnanensis, 135 Mya between T. japonica and Trachytettix bufo, 131.2–271.1 Mya between T. japonica and E. minuta, 153.5–186.0 Mya between E. minuta and M. andensis. The JC69 model was used and the parameters were set to burnin of 100,000, sampfreq of 50 and nsample of 500,000.

Results

Feature and organization

Statistical information on the raw data was presented in Table S2. The mitogenome assembly sequences of five Tetrigoidea species were obtained, including Z. curvispinus, Z. hainanensis, S. melli, E. oculatus and T. yunnana, with the mitogenome of Z. hainanensis being newly announced, while the mitogenome sequences of the other four species were available in the GenBank database. The complete mitogenome sequences were assembled in Z. curvispinus and S. melli, with lengths of 16,874 bp and 16,111 bp, respectively, while the partial mitogenomes of Z. hainanensis, E. oculatus and T. yunnana were obtained with lengths of 8,756 bp, 11,407 bp, and 11,006 bp, respectively. These mitogenomes have been submitted to the GenBank database with accession numbers PQ783659 –PQ783663.

The gene composition and structure of Z. curvispinus and S. melli were shown in Fig. 1, with mitogenomes containing 13 PCGs, two RNAs, 22 tRNAs and one non-coding region (A+T-rich region). A 1,093 bp intergenic region was identified between trnS(ucn) and nad1 in the Z. curvispinus mitogenome, containing tandem repeats with a copy number of 5.6 and a consensus size of 148 bp.

Figure 1 The mitogenome organization of two Tetrigoidea species, including Zhengitettix curvispinus and Scelimena melli.

Compared with the records in the GenBank database, the sequence data of four mitogenomes of studied species (Z. curvispinus, S. melli, E. oculatus and T. yunnana), have been significantly expanded. We annotated the trnI, trnQ and trnM in the newly assembled mitogenome sequences of Z. curvispinus and E. oculatus and compared our sequences to the mitogenomes of Z. curvispinus (MT162544) and E. oculatus (MT162546). The A+T-rich region sequence was obtained and annotated for S. melli in our study. We compared the new mitogenome sequence to the previously sequenced S. melli partial mitogenomes (MT162548 and MW722938) and discussed the differences. Furthermore, we show that the intergenic region between trnS(ucn) and nad1 of the Z. curvispinus mitogenome is polymorphic in length and the newly obtained intergenic region differs from the previously sequenced one (MT162544).

Nucleotide composition

The nucleotide composition and skewness of different mitogenome datasets in Z. curvispinus and S. melli were similar (Fig. S1), with more A bases in the whole mitogenome, PCG-1st (the first position of codons in PCGs) and A+T-rich region, more T bases in PCG, PCG-2nd (the second position of codons in PCGs), PCG-3rd (the third position of codons in PCGs), rRNA and tRNA datasets. In the whole mitogenome of Z. curvispinus, 44.3% A, 15.5% C, 9.1% G and 31.1% T were found, and in S. melli, 41.4% A, 19.4% C, 11.1% G, and 28.1% T were identified. The A+T content of 75.4% in Z. curvispinus and 69.5% in S. melli represented a higher proportion of A+T compared to that of G+C.

A- and C-skew were found in the whole mitogenomes of these two sampled Tetrigoidea species. The PCG dataset showed a tendency towards T- and C-skew. Among the three codon positions of PCGs, A- and G-skew were found in the PCG-1st dataset, T- and C-skew was identified in the PCG-2nd and PCG-3rd datasets. Obvious T-skew was observed in the PCG-2nd dataset, while obvious C-skew was found in the PCG-3rd dataset. T- and G-skew in rRNAs, A- and G-skew in tRNAs, A- and C-skew in the A+T-rich region were identified.

Protein-coding genes

Consistent with other Orthopteran species (Zhang et al., 2013; Chang et al., 2020; Li et al., 2021), nine mitogenomic PCGs were encoded on the major strand (J-strand) and the remaining four PCGs (nad5, nad4, nad4L and nad1) were encoded on the minor strand (N-strand) (Fig. 1). For Z. curvispinus PCGs, four types of initiation codons were identified, including ATA of nad2 and nad3, ATC of cox1 and nad6, ATT of nad4L and nad1, and ATG of the remaining seven PCGs, while three types of termination codons were found in PCGs, including TAG of nad3, incomplete T of cox1, cox3 and nad5, and TAA of the remaining nine PCGs. Four pattern initiation codons were used in S. melli PCGs, including ATC in nad2, cox1, ATT in nad3, nad4L and nad1, TTG in nad6, and ATG in the remaining seven PCGs, while three termination codons were found in PCGs, including TAG in nad3 and cytb, incomplete T in cox1, cox3 and nad5, and TAA of the remaining eight PCGs.

Start and stop codon comparisons of 50 Tetrigoidea mitogenomes showed that PCGs started with six types of initiation codons, including ATA, ATG, ATT, ATC, TTG and GTG (Fig. S2A). ATG of cytb and ATC of cox1 were conserved. PCGs were terminated with four pattern termination codons, including TAA, TAG, TA and T (Fig. S2B). TAA of atp8, atp6 and nad6 were also conserved.

RSCU results in Z. curvispinus and S. melli PCGs showed that the most frequently used codons were UUA(L), UCA(S), CCU(P) and UCU(S) (Fig. 2). The codons with U and A in third position were most frequently used.

Figure 2 The RSCU component of two Tetrigoidea species.

(A) Zhengitettix curvispinus; (B) Scelimena melli.

RRNA genes

For two rRNA genes in the Z. curvispinus and S. melli mitogenomes, rrnL was located between trnL(cun) and trnV, while rrnS was located between trnV and the A+T-rich region (Fig. 1). The length of the rrnL genes was 1,293 bp in the Z. curvispinus and S. melli mitogenomes, while that of rrnS was 789 bp in Z. curvispinus and 778 bp in S. melli, respectively.

TRNA genes

Similar to other Orthopteran mitogenomes (Zhang et al., 2013; Chang et al., 2020; Li et al., 2021), 14 tRNAs were encoded on the J-strand, while the remaining eight tRNAs (trnQ, trnC, trnY, trnF, trnH, trnP, trnL(cun) and trnV) in the whole Z. curvispinus and S. melli mitogenomes were encoded on the N-strand (Fig. 1).

All tRNAs obtained were interspersed in mitogenomes, with sizes ranging from 62 bp (trnD, trnF and trnP) to 72 bp (trnV) in Z. curvispinus and from 62 bp (trnD, trnG and trnF) to 71 bp (trnV) in S. melli. Typical clover secondary structures were found in 22 tRNAs in the Z. curvispinus (Fig. S3). For trnS(agn) in five sampled Tetrigoidea mitogenomes, the secondary structures of the trnS(agn) gene were different, lacking the DHU arm in four species (Fig. S4), but containing the DHU arm in Z. curvispinus with three AU pairs and one UU mismatch (Fig. S3).

A+T-rich region

The non-coding region of the A+T-rich region was located between rrnS and trnI (Fig. 1), with a length of 1,075 bp in Z. curvispinus and 1,498 bp in S. melli. The length of the A+T-rich region varied more in sampled available 27 Tetrigoidea mitogenomes, ranging from 413 bp (Bolivaritettix sikkinensis, KY123120) to 3,290 bp (Paragavialidium hainanense, NC_071831). Tandem repetitive sequences of 59 bp were identified in the Z. curvispinus with a copy number of 16.4 and 213 bp were found in the S. melli with a copy number of 2.8. The length, position, copy number and consensus pattern of tandem repeats (TRs) in the A+T-rich region were different in species of the genus Zhengitettix and Scelimena (Table S3), with more TRs found in Scelimena sp. (OP057410).

Phylogenetic analysis

There was no significant saturation in three datasets (PCG, PCG_rRNA and PCG_rRNA_tRNA) (Fig. S5), with Iss <Iss.c. Phylogenetic analyses using ML and BI methods based on three different mitogenome datasets yielded a consistent topology (Fig. 3 and Fig. S6), except for T. bufo, the clade of Teredorus/Zhengitettix hainanensis/Systolederus, Criotettix japonicus and Bolivaritettix. Bootstrap values (BSs) in ML trees and posterior probabilities (PPs) in BI trees were relatively high at most nodes, including some nodes with strong support of BS = 100 and PP = 1, such as the clade containing Paragavialidium/Scelimena. Six subfamilies (Batrachideinae, Tripetalocerinae, Scelimeninae, Cladonotinae, Tetriginae and Metrodorinae) and two tribes (Criotettigini and Thoradontini) of Tetrigoidea were included in the phylogenetic analyses.

Figure 3 Phylogenetic trees of Tetrigoidea species reconstructed using three mitogenomic datasets.

Note: Node supports from left to right: PCG ML tree, PCG BI tree, PCG_rRNA ML tree, PCG_rRNA BI tree, PCG_rRNA_tRNA ML tree, and PCG_rRNA_RNA BI tree; *, bootstrap support of 100 in all ML trees and Bayesian posterial probability of 1.00 in all BI trees; –, not support in the node.

The Tetrigoidea phylogenetic tree was divided into seven clades (Fig. 3), with clade I including Batrachideinae; clade II including Tripetalocerinae; clade III including majority of species of Scelimeninae; clade IV including one species of Cladonotinae; clade V including partial Tetriginae species, a Scelimeninae and Metrodorinae species; clade VI including a Scelimeninae species and majority of Tetriginae species; clade VII including a Scelimeninae and Cladonotinae species, Criotettigini, majority of Metrodorinae species and Thoradontini. Monophyly of most subfamilies was not supported, but several stable branches existed in the phylogenetic tree, such as clades III, V, VI and the branch containing Mazarredia convexa, Yunnantettix bannaensis and Thoradontini within clade VII.

For Batrachideinae (clade I), Tripetalocerinae (clade II) and Criotettigini (within clade VII), as fewer species were included in this study, their monophyly was not considered. However, their phylogenetic status was stable, with Batrachideinae and Tripetalocerinae forming the basal position of Tetrigoidea with relatively high BS and PP values in clades I and II (Fig. 3). For Criotettigini, C. japonicus formed the basal position of a complex clade of Metrodorinae/Thoradontini/Cladonotinae in clade VII in the PCG trees (BS = 100 and PP = 1) and the PCG_rRNA ML tree (BS = 100) (Fig. 3), while C. japonicus was at the basal position of the genus Bolivaritettix also in clade VII in the remaining phylogenetic trees (PCG_rRNA BI trees and PCG_rRNA_tRNA trees) (Fig. S6).

For Scelimeninae, monophyly was not supported, with Zhengitettix (clades V and VI) and Falconius longicornis (clade VII) not clustered with the main clade containing Paragavialidium and Scelimena (clade III). Two tribes within Scelimeninae were included in the phylogenetic analyses, including Scelimenini (Scelimena and F. longicornis) and Discotettigini (Paragavialidium). Scelimenini was also not monophyletic, with F. longicornis clustering in the basal position of the clade containing Thoradontini, Y. bannaensis, M. convexa, Metrodorinae and C. japonicus, with BS = 99 in ML trees and PP = 1 in BI trees. The genus Scelimena formed a sister group with Paragavialidium (Discotettigini) in clade III, with BS = 100 in ML trees and PP = 1 in BI trees, and this clade (III) was located relatively basal to the Tetrigoidea.

Among Scelimeninae, Z. hainanensis (clade V) was not clustered together with Z. curvispinus (clade VI), among which Z. hainanensis had a closer relationship with S. spicupennis (Metrodorinae) with high BS and PP values (such as BS = 91 and PP = 1 in PCG trees), and then this clade clustered with the genus Teredorus in clade V. Z. curvispinus formed a basal position of the main Tetriginae clade including Tetrix, A. yunnanensis, Formosatettix qinlingensis, Exothotettix guangxiensis, Euparatettix, Coptotettix and Ergatettix in clade VI, also with high BS and PP values.

To further explore the phylogenetic status of Z. hainanensis, the mitogenome sequence of another individual of Z. hainanensis was also sequenced, assembled and annotated. The p-distance was calculated among closely related species based on the cox1 gene, including five Teredorus species, two Z. hainanensis mitogenomes, two S. spicupennis mitogenomes and two Z. curvispinus mitogenomes (Fig. S7). The result showed that the p-distance between two Z. hainanensis individuals had the lowest value, relatively lower for Z. hainanensis compared to T. hainanensis and S. spicupennis, but relatively higher for Z. hainanensis compared to Z. curvispinus. The p-distance results were also consistent with the phylogenetic relationship.

In addition, 24 Tetrigoidea mitogenomes (including the newly published record of Z. transpicula) and two outgroups formed another dataset to investigate the phylogenetic status of the genus Zhengitettix (Table S1), using the PCG dataset, ML and BI methods. The phylogenetic trees showed consistent topologies, with Z. transpicula forming a basal position of the clade containing Z. hainanensis, S. spicupennis and T. hainanensis (((Z. hainanensis, S. spicupennis), T. hainanensis), (Z. transpicula)), but not clustering with Z. curvispinus (Fig. S8). Therefore, the monophyly of Zhengitettix was also not supported.

For Tetriginae, it was also not monophyletic, with the genus Teredorus species in clade V not clustered with other Tetriginae species in clade VI. The monophyly of the genus Teredorus was not recovered, and this genus had a closer relationship with Z. hainanensis and S. spicupennis.

For Metrodorinae, the non-monophyly of this subfamily was supported, with the genus Systolederus (clade V) and Bolivaritettix (clade VII) and M. convexa (clade VII) not forming one clade. Among them, the genus Bolivaritettix formed the basal position of the clade of tribe Thoradontini, M. convexa and Y. bannaensis in the PCG trees, PCG+rRNA ML tree (Fig. 3), while the Bolivaritettix clustered with C. japonicus, with PP = 0.78 in the PCG+rRNA BI tree, BS = 54 in the PCG+rRNA+tRNA ML tree, and PP = 1 in the PCG+rRNA+tRNA BI tree (Fig. S6). M. convexa formed a sister group with Loxilobus prominenoculus, with BS = 100 and PP = 1 in all trees.

As for Cladonotinae, its non-monophyly was supported in all trees, with T. bufo (clade IV) and Y. bannaensis (clade VII) not clustered together. T. bufo formed a basal position of a complex clade containing Thoradontini, Y. bannaensis, Metrodorinae, C. japonicus, Scelimeninae and Tetriginae in the PCG and PCG_rRNA trees (Fig. 3), whereas T. bufo was placed at the basal position of a clade containing Teredorus, Z. hainanensis and S. spicupennis in PCG_rRNA_tRNA trees, with BS = 49 and PP = 0.64 (Fig. S6). For Thoradontini, this tribe was not a monophyletic group, with Thoradonta (clade VII) species clustering with Y. bannaensis (Cladonotinae) in all phylogenetic trees (Fig. 3). Within the genus Thoradonta, the phylogenetic relationship of ((T. nodulosa, T. obtusilobata), T. yunnana) was recovered.

Divergence time

The divergence times showed that the Tetrigoidea split with the outgroup (Tridactyloidea) at 214.7 Mya during the Triassic (Fig. 4). Tetrigoidea genera diverged over relatively long periods, ranging from 193.9 Mya (Saussurella with other Tetrigoidea genera) in the Jurassic to 6.2 Mya (between Tetrix and Alulatettix) of the Neogene. The divergence times of most Tetrigoidea genera lie in the Cretaceous, such as Paragavialidium and Scelimena of the Scelimeninae.

Figure 4 Divergence time result of Tetrigoidea species based on the PCG dataset.

For three subfamilies, Batrachideinae, S. borneensis represented the most original taxon of Tetrigoidea and diverged first at 193.9 Mya, followed by T. tonkinensis (Tripetalocerinae) with a divergence time of 169.65 Mya also during the Jurassic (Fig. 4). For Scelimeninae, the main clade containing Paragavialidium and Scelimena diverged from other Tetrigoidea species at 147.62 Mya during the Cretaceous. Z. curvispinus diverged with the main Tetriginae species at 106.21 Mya, Z. hainanensis diverged with S. spicupennis (Metrodorinae) at 48.27 Mya during the Paleogene, and F. longicornis diverged at 111.55 Mya. The intraspecific divergence time of some Scelimeninae species was also long, such as from 0.03 Mya to 87.91 Mya for Scelimena species.

For the other three subfamilies, Cladonotinae, the divergence time between T. bufo and other Tetrigoidea species was 139.28 Mya during the Cretaceous, while Y. bannaensis split with Thoradonta (Thoradontini) at 47.67 Mya during the Paleogene. For Tetriginae, the genus Teredorus split with other Tetrigoidea species at 131.49 Mya during the Cretaceous, representing a relatively original genus within Tetriginae, and the main Tetriginae clade split with Z. curvispinus also during the Cretaceous. The time of divergence of genera within Tetriginae was maintained over a long period. For Metrodorinae, the divergence time of S. spicupennis, Bolivaritettix and M. convexa was 48.27 Mya, 85.48 Mya and 41.77 Mya, respectively.

For two tribes, Criotettigini, C. japonicus split during the Cretaceous at 95.57 Mya. Within Thoradontini, the divergence time of E. oculatus, L. prominenoculus and Thoradonta was 68.13 Mya, 41.77 Mya and 47.67 Mya, respectively.

Discussion

Mitogenomic structure

In addition to two complete mitogenomes, we also reported three partial mitogenomes in this study. Similar to some previous Tetrigoidea studies, the incomplete mitogenomes lacked the rrnS gene and the A+T-rich region, suggesting that some Tetrigoidea mitogenomes were difficult to assemble using either Sanger sequencing or high-throughput sequencing platforms. The gene order of trnD-trnK was found in the five Tetrigoidea species obtained, which was consistent with other Tetrigoidea mitogenomes (Li et al., 2021). This gene order was also found in all Acrididea species (Gaugel et al., 2023), but was different from that of the insect ancestor (gene order of trnK-trnD) (Song et al., 2015; Gaugel et al., 2023).

The large intergenic region found between trnS(ucn) and nad1 in Z. curvispinus was similar to the previous study (Li et al., 2021), with a length of 945 bp and a tandem repeat sequence of 5 ×148 bp also in Z. curvispinus (MT162544), while 270 bp in Z. transpicula (PQ869509) but without tandem repeats. This long intergenic region has been found in some other Orthoptera species, such as F. longicornis (Li et al., 2021) and Sinochlora longifissa (Liu et al., 2013). This high A+T content in the whole mitogenome found in Z. curvispinus and S. melli was common in Orthoptera mitogenome sequences (Liu et al., 2013). The higher A+T content was an important feature in insect mitogenome sequences, and a previous study showed that the C→T mutation in insect mitogenomes resulted in the high A+T content of mitochondrial genes (The Honeybee Genome Sequencing Consortium, 2006). The PCG dataset showed a T-skew and C-skew tendency, which was consistent with other Tetrigoidea mitogenomes (Li et al., 2021).

Besides the typical ATN initiation codons, TTG and GTG were also found in Tetrigoidea mitogenomes (Li et al., 2021). The incomplete stop codon T was mainly found in cox1, nad5 and cox3 and can generate functional terminal codons via post-transcriptional polyadenylation (Ojala, Montoya & Attardi, 1981). The third position codons ending in U and A were the most commonly used, which was also found in other Tetrigoidea mitogenomes (Li et al., 2021).

Previous studies showed that the trnS(agn) gene lacked the DHU arm, a common feature observed in some other Caelifera species (Zhang et al., 2013; Wang et al., 2023). However, the trnS(agn) gene in Orthopteran species did not always lack the DHU arm. For example, a GC match pair was found in Tarragoilus diuturnus (Orthoptera, Ensifera, Hagloidea) (Zhou, Shi & Zhao, 2014), while AU and GU pairs were identified in Macromotettixoides (Orthoptera, Caelifera, Tetrigoidea) (Luo, Zhang & Deng, 2024). Several stem structures of tRNAs in Z. curvispinus were conserved compared to other Tetrigoidea species. For example, the acceptor arm of trnI, trnK and trnT, the DHU arm of trnI, trnQ, trnM, trnW, trnC, trnL(uur), trnD, trnK, trnG, trnA, trnE, trnH, trnT, trnP and trnL(cun), the anticodon arm of trnG, trnN, trnE and trnV, the TψC arm of trnS(ucn) in the Z. curvispinus was consistent with that of published Macromotettixoides species (Luo, Zhang & Deng, 2024).

A previous study indicated that the control region of insects was variable in both size and nucleotide sequence (Zhang & Hweitt, 1997), which was also reflected in the genus Zhengitettix, with a length of 270 bp in Z. transpicula, 1,075 bp in Z. curvispinus (PQ783659) and 998 bp in Z. curvispinus (MT162544). Insect control region may contain tandem repeats (Zhang & Hweitt, 1997), and this feature was also reflected in species of the genera Zhengitettix and Scelimena. For example, more TRs were found in Scelimena sp. (OP057410), probably due to the much longer length of the A+T-rich region. Topological structures of TRs have been found in some insects, such as a super-cruciform structure in Nasonia vitripennis (Lin et al., 2021). The tandem repeats appeared to have undergone concerted evolution, and copy number variation showed a high mutation rate (Zhang & Hweitt, 1997).

Nuclear mitochondrial DNA sequences (NUMTs) have been identified in many insects (Wang et al., 2020; Pereira et al., 2021; Hebert, Bock & Prosser, 2023; Liu et al., 2024). For example, Hebert, Bock & Prosser (2023) detected nearly 10,000 cox1 NUMTs (≥100 bp) in the genomes of 1,002 insect species. At the genome level, the number of NUMT insertions was significantly positively correlated with the transposable element (TE) content of Orthoptera genomes (Liu et al., 2024), and the inserted NUMTs may also be influenced by TEs in the neighbors of fig wasps (Hymenoptera) (Wang et al., 2020). In addition, the NUMT content varied widely in fig wasps, and NUMT bursts existed in some Hymenoptera species or lineages; the large number of NUMTs might be related to the large genomes of Hymenoptera species, and NUMTs tended to be inserted in unstable regions of the genomes (Wang et al., 2020). Furthermore, the insect orders with the largest genome sizes had the highest number of NUMTs, but there was considerable variation among their component lineages (Hebert, Bock & Prosser, 2023). At the transcriptome level, Liu et al. (2024) analyzed the NUMTs of Orthoptera insects and proposed the potential mechanism of NUMT integration, that mitochondrial transcripts are reverse transcribed into double-stranded DNA (dsDNA) and then integrated into genomes. At both the genome and transcriptome level, Pereira et al. (2021) used Chorthippus parallelus, with a genome size of ∼13 Gb, to identify NUMTs and found that the mitogenome region containing the cox1 gene had disproportionately higher diversity and coverage than the other region, consistent with multiple insertions of the region into the nuclear genome.

Phylogeny and evolution

The phylogenetic trees of Tetrigoidea reconstructed in this study were stable, and several phylogenetic relationships were consistent with previous studies, such as S. spicupennis clustering near the clade Tetriginae (Li et al., 2021). Within Scelimeninae, the genus Scelimena formed a sister group with Paragavialidium, which phylogenetic result was consistent with the previous study based on mitochondrial genes combined with nuclear segment (Chen et al., 2018b; Wei et al., 2023). And this clade (III) was located at the relatively basal position of Tetrigoidea, which was different from that of Wei et al. (2023) and Guan, Huang & Deng (2024), but similar to that of Luo, Zhang & Deng (2024).

The monophyly of most subfamilies of Tetrigoidea was not supported, and these non-monophyly were found in previously studied subfamilies, such as Scelimeninae and Cladonotinae. The non-monophyly of Scelimeninae was reconstructed based on mitogenomic datasets (Li et al., 2021; Qin et al., 2023; Wei et al., 2023; Luo, Zhang & Deng, 2024), mitochondrial genes, nuclear fragments and morphological characters (Adžić et al., 2020). The classification of Scelimennae, Metrodorinae and Tetriginae is usually distinguished by the shape of the posterior angles of the pronotal lateral lobes as a morphological character, but this shape is sometimes difficult to assess accurately. For example, the genus Systolederus (Metrodorinae) and Teredorus (Tetriginae) were confirmed to be the same genus (Li et al., 2021). And the non-monophyly of Cladonotinae was consistent with the results from mitogenomic data (Li et al., 2021; Li, Liu & Lin, 2021). However, Tetriginae was monophyletic in the previous study (Li et al., 2021; Wei et al., 2023), which was different from our phylogenetic inferences, probably due to the relatively larger Tetrigoidea species sampled in our study.

Some branches were confused among some species, genera and subfamilies. Based on the phylogenetic trees, the phylogenetic status of some Tetrigoidea species, such as T. bufo and C. japonicus, needed to be further investigated with more extensive sampling. The phylogeny within the genus Thoradonta was consistent with the study by Wei et al. (2023). However, this genus was clustered with Yunnantettix (Y. bannaensis) in our study, rather than with Loxilobus (L. prominenoculus) in the study of Wei et al. (2023). This difference may be due to different sampling strategies. Moreover, Z. hainanensis is the type species of the genus Zhengitettix (Orthoptera Species File v.5.0), and most studies on this genus have focused on morphological classification (Deng, Zheng & Wei, 2010; Zheng, Zeng & Ou, 2010; Storozhenko, 2013; Chen & Deng, 2022). Phylogenetic trees showed that Z. hainanensis and Z. transpicula were clustered with S. spicupennis and T. hainanensis separately. All of these species shared some morphological taxonomic characters, such as small size and narrow vertex, suggesting that they may have a closer phylogenetic relationship. The phylogenetic status of Z. curvispinus was different, as it formed a basal position of the main Tetriginae clade in our study, while it formed a basal position of Tetrigoidea, including species of Tetriginae, Metrodorinae, Cladonotinae, Scelimeninae, Thoradontini and Criotettigini (Li et al., 2021), located in a basal position of a clade containing Metrodorinae, Thoradontini and Criotettigini (Luo, Zhang & Deng, 2024), grouped with Xistra strictivertex (Metrodorinae) (Wang, 2022), and formed a basal position of some Scelimeninae species, including F. longicornis, Paragavialidium sichuanense and S. melli (Wei et al., 2023). The characters of small size and narrow vertex were also present in X. strictivertex. All these analyses showed that the species of Zhengitettix (Scelimeninae) were a complex taxa and should be concerned in phylogeny, and its phylogenetic relationship and non-monophyly should also be further investigated from morphological classification and molecular genetics, using a larger sample size and an appropriate phylogenetic strategy.

The applicability of some morphological characters to a particular family, genus and species classification in Tetrigoidea needs further study. Species homonymy and transfers from one genus to another have occurred frequently. For example, Long et al. (2023) reported 23 new synonyms of T. japonica. These foundings showed that hybridization may have occurred within Tetrigoidea species, and that molecular data, especially genome sequences, were needed to analyze their speciation. Furthermore, the morphological classification of subfamilies might be problematic, and some characters are not sufficient to distinguish the existing subfamilies, leading to confusion in the classification and monophyly of these taxa. Therefore, more sampling is needed to verify whether the phylogenetic results are caused by the sampling strategy, and we will further verify the morphological characters of the Tetrigoidea species based on molecular data with large samples.

The divergence time supported that the Tetrigoidea species was one of the more ancient lineages (Song et al., 2015). The divergence results in this study were later than those in previous study (Guan, Huang & Deng, 2024). S. borneensis (Batrachideinae) represented the most basal taxon of the Tetrigoidea, with a similar divergence time of 201.35 Mya (Guan, Huang & Deng, 2024), followed by T. tonkinensis (Tripetalocerinae). Most genera of Tetrigoidea diverged during the Cretaceous, a period of great importance for the evolution of these taxa.

Conclusions

Five Tetrigoidea mitogenomes were sequenced and genomic features were analyzed. A+T content was identified in the mitogenomes, with A- and C-skew. ATG and TAA were the most common initiation and termination codons. Several non-monophyly subfamilies and tribes were found, such as Scelimeninae and Thoradontini. Tetrigoidea species were relatively original taxa in Orthoptera, with Batrachideinae and Tripetalocerinae diverging earlier. Our results provide a valuable resource for further research on the mitogenomic character, phylogeny and evolution of Tetrigoidea.

Supplemental Information

Supplemental Information 1 Supplemental information

We would like to special thanks to Prof. Weian Deng of the Guangxi Normal University for collecting and identifying the specimens.

Additional Information and Declarations

Competing Interests

Author Contributions

DNA Deposition

Data Availability

The authors declare there are no competing interests.

Xuejuan Li conceived and designed the experiments, performed the experiments, analyzed the data, prepared figures and/or tables, authored or reviewed drafts of the article, and approved the final draft.

Wenli Dou performed the experiments, prepared figures and/or tables, authored or reviewed drafts of the article, and approved the final draft.

Liliang Lin conceived and designed the experiments, performed the experiments, analyzed the data, prepared figures and/or tables, authored or reviewed drafts of the article, and approved the final draft.

The following information was supplied regarding the deposition of DNA sequences:

The mitogenome sequence data are available at GenBank: PQ783659 –PQ783663.

The following information was supplied regarding data availability:

The mitogenome sequence data are available at GenBank: PQ783659 –PQ783663. Other data are available in the Supplementary File.

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
