# Peer review of "Mitogenomic phylogeny of Tetrigoidea (Insecta, Orthoptera), with a focus on the genus Zhengitettix"

_PeerJ, doi:10.7717/peerj.19521_

## Round 0.1 · original submission · Major Revisions

Dear Authors,

Thank you for submitting your manuscript to PeerJ. After a thorough review by three independent reviewers, it has been determined that significant improvements are necessary.

I strongly encourage you to carefully proofread the manuscript to enhance its clarity and readability. Additionally, the Materials and Methods and Results sections require substantial revisions to strengthen the overall quality of the work. Given the extent of these necessary improvements, major revisions are required before the manuscript can be reconsidered.

I look forward to receiving your revised version.

Best regards,
Armando Sunny

Reviewer 1 ·

Basic reporting

I reviewed the manuscript entitled “The mitogenomes of five Tetrigoidea species, providing new insight into phylogenetic implications”. In this study, the authors performed parametric phylogenetic reconstructions using different mitogenomic datasets of various Tetrigoidea species to assess phylogenetic relationships of different tetrigoid taxa as well as test the monophyly of some of them. The authors also estimated time of divergence of main lineages within the group.
I do not have major concerns regarding the methods used for phylogenetic reconstruction. However, various aspects regarding wet lab procedures, raw data processing, dating analyses, taxon sampling strategy as well as discussion, require further clarification and improvement.

Experimental design

The authors reported assembling and annotating five mitogenomes of tetrigoids “… including Z. curvispinus, Z. hainanensis, S. melli, E. oculatus and T. yunnana) and Z. hainanensis being newly sequenced…”. This means that only one of the four generated genomes corresponds to a species without previous mitogenomic data, while the remaining four represent newly sequenced mitogenomes of species for which mitogenomic data was already available in Genbank. This is why the phylogenetic trees include multiple representatives of the same species. Subsequently, several parts of the manuscript, including tittle require revision to reflect this more accurately.
It is unclear how the calibration points were set. The authors mention using four calibration points, one of which was placed “between” Tetrix japonica and Ellipes minuta, however, these taxa are not closely related, as E. minuta is one of the two outgroup taxa used. Similarly, another calibration point was placed “between” T. japonica and Trachytettix bufo, but same, these are not closely related taxa. It is not clear - at least as explained- how it is accurate to place a calibration point in the “divergence” of two taxa that are not sister or not even closely related in the phylogenetic hypothesis obtained. Further clarification and justification are need for these calibration choices. Additionally, there is no mention of which clock model was used for divergence estimation in MCMCTREE.

Validity of the findings

In the introduction, the authors mention that previous studies on Tetrigoidea left some questions unresolved, such as the monophyly of Scelimeninae. This explain why the authors generated additional mitogenomes of the same species—to assess by increasing taxon sampling whether this subfamily is monophyletic. However, this sampling strategy is not ideal, as it does not incorporate a broader spectrum of known Scelimeninae diversity. Instead, it only adds terminal taxa in the exact same position in the phylogeny by including multiple mitogenomes of the exact same species. As a result, this study recovered a similar result as reported in previous studies (performed by the same authors of the present contribution), with Zhengitettix curvispinus failing to cluster with remaining Scelimeninae, rendering the subfamily as non-monophyletic.
It is evident that mitogenomic data is providing evidence that some groups within Tetrigoidea are not monophyletic based on this source of data. However, the manuscript does not adequately discuss the implication of the non-monophyly of Scelimeninae (or other groups that were also found not to be monophyletic). The authors could have strengthened their discussion by finding other sources of evidence that could support this result, such as mentioning some morphological features that differentiate Scelimena, Zhengitettix and Falconius. Similarly for other subfamilies and tribes that were recovered as non-monophyletic and not only

Additional comments

Here some comments to consider.
- Please provide more details in how gDNA was extracted, e.g., if DNA was extracted from legs, whole specimens, other strategy.
- Replace “Total genomic DNA was extracted…” for “Genomic DNA was extracted...”
- What was the final volume when resuspended DNA extractions?
- What was input DNA used per sample for library prep
- Did the auuthors use sonicator for DNA fragmentation? Please indicate model.
- What kit for library prep was used?
- Wat was total raw data (reads) generated per sample after sequencing?
- After trimming, how many reads per sample were used for assembly?
- When describe recovered topology, I deeply encourage the authors do not refer to each obtained clade as “Phylogeny of Scelimeninae, Phylogeny of the genus Teredorus, etc.” Obtained results are not isolated, there was not a phylogenetic reconstruction per taxonomic group within Tetrigidae. It is a single phylogenetic hypothesis of the major taxon Tetrigidae (Tegrigoidea).

Reviewer 2 ·

Basic reporting

1) Unfortunately, English is not my native language and it's hard for me to give advice on how to improve it, but some phrases are difficult to understand. The English language should be improved to ensure that an international audience can clearly understand your text. Some examples where the language could be improved include:
lines 14-16 - something is missing in the last part of the sentence;
lines 24, 369 - the word “original” is not appropriate in this context;
line 67 – “evolutionary” should be changed to “evolution”;
lines 105-106 – I didn't understand the meaning. What was done and in what order?;
lines 157, 159 – “majority strand” and “minority strand” should be changed to “major strand” and “minor strand”;
lines 173-174 - the phrase needs to be rewritten. For example, “The codons ending in U and A were most frequently used” or “The codons with U and A in third position were most frequently used”;
lines 208, 210, 212 – “major species” is better to change to “majority of species”
I think, there are some other places that need to be improved. I suggest you have a colleague who is proficient in English and familiar with the subject matter review your manuscript, or contact a professional editing service

2) It would be good if you explained your choice of species for study in introduction. As I can see, mt-genomes for four species from five have already been sequenced and analyzed. a) What is known about the biology of these species? What role does behavioral isolation play in speciation of these species? Is hybridization with subsequent introgression of mtDNA possible? This information will allow you to predict the expected range of genetic distances at the intraspecific level and the level of closely related species. b) What are the difficulties with diagnostics of these species? What is the probability of an incorrect identification? Are there any characteristic differences between the genera? Can a species be incorrectly assigned to a certain genus when described? Molecular genetic analysis allows you to resolve many of these issues. In this case, the choice of species will look more justified.

3) In my opinion, you have not discussed the mtDNA Control Region structure and interspecific differences of your species fully enough. Tandem repeats in the mtDNA CR are common, but I have never seen mitochondrial repeats also found in the nuclear genome, as in the case of Z. curvispinus. I have found homologous repeats in chromosome 4 of Zhengitettix transpicula, as well as in the nuclear genomes of some other invertebrates. Could you please provide your opinion on the origin of these repeats in the discussion?

4) I think, that in the case of non-monophyly of genus, it is not enough to say that more samples are needed. It is necessary to indicate in the discussion your opinion on what the reasons may be. For example, perhaps a revision of some species of the genus is required, or may the genus be paraphyletic and another young genera have formed within it? Or is this a sequencing artifact associated with the inclusion of nuclear copies of mt genes (NUMTs) into mt-genome assembly? The most problematic genus in you study is Zhengitettix. It would be good if you if you include the references on the history of the genus description and the key to species to your reference list. For example: Storozhenko, S.Y. (2013) New and little-known species of the genus Zhengitettix (Orthoptera: Tetrigidae: Scelimeninae) from Southeast Asia. Zoosystematica Rossica, 22(2), 204–223.; or Zheng, Z.-M., Zeng, H.-H. & Ou, X.-H. (2010) A review of the genus Zhengitettix Liang (Orthoptera: Tetrigoidea) with description of one new species. Acta Entomologica Sinica, 53(10), 1153–1156.; or Deng W.A., Zheng Z.M. & Wie S.Z. 2010. A taxonomic study on the genus Zhengitettix Liang (Orthoptera, Tetrigoidea, Scelimenidae). Acta Zootaxonomica Sinica, 35(1): 46–48. It would also be good to point out that Z. hainanensis is the type species of the genus. I think that in the controversial situation with the non-monophyly of Zhengitettix it would be better to include in the phylogenetic constructions the sequences of Zhengitettix transpicula mt-genome, which is already deposited in Genbank, in your phylogenetic reconstructions.

Experimental design

1) Could you please provide more details in the Materials and methods on how the PCG, PCG+rRNA, PCG+rRNA+tRNA trees were constructed, when some of the mt-genomes you sequenced do not contain some tRNA, and the mt-genome of Z. Hainanensis also does not contain rRNA genes and some PCG?

2) Please explain in the Materials and methods how do you avoid contamination of your assembly with NUMT sequences.

Validity of the findings

1) I found at least three articles that provide data on the structure, base composition and codon usage of mt-genomes of four from five species you analyzed: https://doi.org/10.7717/peerj.10523 - for Zhengitettix curvispinus, Scelimena melli and Eucriotettix oculatus; https://doi.org/10.1080/23802359.2021.1978887 (it would be better if you included this item into the reference list) - for Scelimena melli; and DOI: 10.3969/j.issn.2095-1191.2023.05.004 (it should be included too, I think) – for Thoradonta yunnana. Could you discuss in details what new results you are able to show regarding these three studies?

Additional comments

The study provides valuable information about the structure and variability of mtDNA in the Tetrigoidea superfamily. This is especially important since there are few molecular data available for this superfamily. In recent years many articles of this type have been published in a variety of journals of phylogenetic and evolutionary profile, so this study can also be published taking into account the comments given above.

Reviewer 3 ·

Basic reporting

I primarily suggest incorporating the taxonomic classification of the suborder and order in the title to enhance clarity and facilitate future searches in online databases: “The mitogenomes of five Tetrigoidea (Caelifera: Orthoptera) species, providing a new insight into phylogenetic implications”

Your introduction could benefit from additional details. I kindly suggest expanding the description of Caelifera mitogenomes, further exploring their evolution and characteristics within the suborder.

Lines 34–35: I recommend adjusting the order of the clade indication. Currently, it is written as "... belonging to Orthoptera, Caelifera." It would be clearer to state "... belonging to Caelifera, Orthoptera."

In the Results section, under Nucleotide Composition, I suggest clarifying the terminology PCG-1st, PCG-2nd, and PCG-3rd, as these terms appear only in this section and may cause uncertainty regarding the datasets to which they refer.

Lines 157 and 181: It might be helpful to specify which Orthoptera species the results for PCGs are consistent with—whether they align with the species used in your database or with the clade as a whole.

In the Results section, I suggest simplifying the subtitles under Phylogenetic Analysis by including only the names of the highlighted clades, which would improve readability and reduce redundancy. For example: "Phylogeny of the Batrachideinae, Tripetalocerinae, and Criotettigini" to "Batrachideinae, Tripetalocerinae, and Criotettigini"; "Phylogeny of the Scelimeninae" to "Scelimeninae" and so on

Lines 220–223: This passage could be revised for greater clarity and fluency.

Lines 277–283: I recommend rewording this section to avoid repetition of certain terms.

Line 225: It would be beneficial to specify the clades in which Zhengitettix and Falconius longicornis were positioned. For instance: "...with Zhengitettix (clades V and VI) and Falconius longicornis (clade VII) not clustered..."

Lines 228–231: To enhance readability, I suggest rewording this sentence: "Scelimenini was also not monophyletic, with F. longicornis clustering in the basal position of the clade containing Thoradontini, Y. bannaensis, M. convexa, Metrodorinae, and C. japonicus, with BS=99 in ML trees and PP=1 in BI trees."

Line 311: It would be valuable to mention specific Orthoptera species that exhibit a high A+T content.

Figure 1: I suggest removing the word "including" to improve readability.

Figure 4: Some node values overlap with species names. Adjusting their positioning could make the information visually clearer.

Figure S1: I recommend ensuring consistency in the formatting of scientific names. For example, if Zhengitettix hainanensis is written in full, the same format should be applied to S. melli.

Table S1: It would be helpful to visually highlight the newly generated data from your study. This could be achieved by changing the font color or adding a background highlight to distinguish these entries.

Experimental design

Lines 89–90: To facilitate the replication of analyses, it would be important to mention the reference species used for genome assembly and annotation.

I recommend specifying the software programs used for image production and editing so that future studies can utilize the same tools.

A useful analysis that could be applied to enhance the robustness of phylogenetic data is the saturation test, which evaluates substitution rates in sequences. The accumulation of substitutions can introduce noise into phylogenetic reconstructions, potentially obscuring the true evolutionary history.

Validity of the findings

No comment

Additional comments

Dear Authors,

I hope this message finds you well. I have carefully reviewed your manuscript, “The mitogenomes of five Tetrigoidea species, providing a new insight into phylogenetic implications,” and I sincerely appreciate the effort and significance of your research.
With this in mind, I have provided some suggestions that I believe will enhance the clarity and fluency of the manuscript. I am confident that these revisions will further improve the quality of your work and strengthen its contribution to the understanding of Orthoptera.

---

## Round 0.2 · Minor Revisions

Dear Authors,

I hope this message finds you well. I would like to begin by congratulating you on the revisions you have made to your manuscript titled "Mitogenomic Phylogeny of Tetrigoidea (Insecta, Orthoptera), with a Focus on the Genus Zhengitettix." The changes made have significantly improved the manuscript, and your efforts are much appreciated.

However, after a thorough review, I would like to suggest a few minor revisions that will further enhance the clarity and quality of your work. These revisions are the last step toward making the manuscript ready for publication in PeerJ. Once these small adjustments are made, I am confident that the manuscript will be suitable for acceptance.

I look forward to receiving the revised manuscript.

Best regards,
Armando Sunny

Reviewer 2 ·

Basic reporting

Dear authors, I suggest making a few small corrections.
Line 23 – the word “ancient” is also not very good here. May be “that Tetrigoidea is one of the basal branch of Orthoptera,..”?
Lines 114-116 – I don’t understand the sentence. What stop codons are you talking about? PGS has only one stop codon – at the end of sequence. The sentence needs to be reworded.
Line 117 – it is better to change “lacking” to “lacked” or “lost”.
Lines 151-157 – “Compared with the records in the GenBank database, the sequence data of four mitogenomes of studied species (Z. curvispinus, S. melli, E. oculatus and T. yunnana), have been significantly expanded. We annotated the trnI, trnQ and trnM in the newly assembled mitogenome sequences of Z. curvispinus and E. oculatus and compared our sequences to the mitogenomes of Z. curvispinus (MT162544) and E. oculatus (MT162546). The A+T-rich region sequence was obtained and annotated in for S. melli in our study. We compared the new mitogenome sequence to the previously sequenced S. melli partial mitogenomes (MT162548 and MW722938) and discussed the differences. Furthermore, we show that the intergenic region between trnS(ucn) and nad1 of the Z. curvispinus mitogenome is polymorphic in length and the newly obtained intergenic region differs from the previously sequenced one (MT162544).”
Line 343 – “the insect control region of insects” – repetition.
Line 361 – Change “in previous studies” to “in previously studied subfamilies”
Lines 323-324 - The higher A+T content in mtDNA is an important feature not only for Orthoptera, but also for all insects.
Line 418 – what is “partial species”?
Lines 373-374 - for the genus Zhengitettix, Z. hainanensis is the 374 type species of the genus Zhengitettix – repetition
Lines 376-378 – it is better to reword.

Experimental design

- I think you may briefly discuss the problem of NUMTs. It may be interesting for researchers studying other groups of Orthoptera, such as Gomphocerinae with very large genomes and hundreds of copies of mtDNA sequences in their nuclear genomes. (DOI: 10.1111/jzs.12446)

Validity of the findings

- I found that the mtDNA sequences of S. melli from this study (PQ783660) and from previous study (MW722938) have not nucleotide differences except of newly sequenced A+T rich region. Is this the same specimen?
- I think it would be good to discuss the differences of your phylogenetic implications from previously published results (https://doi.org/10.3969/j.issn.2095-1191.2023.05.004

Additional comments

no comment

Reviewer 3 ·

Basic reporting

Lines 405-408: To improve readability, I suggest rewording this sentence as follows: “Moreover, Z. hainanensis is the type species of the genus Zhengitettix (Orthoptera Species File v.5.0), and most studies on this genus have focused on morphological classification (Deng, Zheng & Wei, 2010; Zheng, Zeng & Ou, 2010; Storozhenko, 2013; Chen & Deng, 2022).”

Lines 436–437: It might be more accurate to refer to S. borneensis as the 'most basal taxon' rather than 'most original.'

Experimental design

No comment

Validity of the findings

No comment

Additional comments

The authors have adequately addressed the suggested revisions, making the manuscript more comprehensive. It represents a valuable contribution to the study of Orthoptera systematics and evolution.

---

## Round 0.3 · accepted · Accept

Dear Dr. Li,

Thank you for submitting your manuscript entitled “Mitogenomic phylogeny of Tetrigoidea (Insecta, Orthoptera), with a focus on the genus Zhengitettix” to PeerJ. I would like to express my appreciation for the revisions you made in response to the reviewers’ comments.

After careful consideration and evaluation of the revised version, I am pleased to inform you that your manuscript has been accepted for publication. The reviewers and I found the study to be a valuable contribution to the field of insect phylogenetics, with robust methodology and clear presentation of results.

Your article will now proceed to the production phase. You will be contacted shortly by our editorial office regarding the final steps prior to publication.

Congratulations, and thank you for choosing PeerJ as the venue for your work. We look forward to seeing your article published soon.

Best regards,
Dr. Armando Sunny

Reviewer 2 ·

Basic reporting

Dear authors, one more small comment

Line 23 - Replace "branch" with "branches"

Lines 356-372 - Sorry for insisting on introducing the text about NUMTs. It turned out to be too detailed. I think it would be enough to say that the NUMTs may be a problem for mtDNA sequences analyzis, especially in the case of Orthoptera with large genomes. The genomes of some Gomphocerinae reach the size ~13 Gb and contain hundreds of NUMTs copies (Pereira et al., 2021). However, the genomes of Tetrigoidea are much smaller. The genome of Zhengitettix transpicula is ~1 Gb (doi: 10.3390/insects15040223) and contain only a few copies of NUMTs (this may be checked by BLAST the mtDNA on the genome of Zhengitettix transpicula). Thus, NUMTs cannot significantly complicate the assembling of mitogenome of Tetrigoidea.

Experimental design

no comment

Validity of the findings

no comment

Additional comments

no comment